# Social Determinants of Obesity and Stunting among Brazilian Adolescents: A Multilevel Analysis

**DOI:** 10.3390/nu14112334

**Published:** 2022-06-02

**Authors:** Diôgo Vale, Maria Eduarda da Costa Andrade, Natalie Marinho Dantas, Ricardo Andrade Bezerra, Clélia de Oliveira Lyra, Angelo Giuseppe Roncalli da Costa Oliveira

**Affiliations:** 1Postgraduate Program in Collective Health, Federal University of Rio Grande do Norte, Natal 59056-000, RN, Brazil; clelia.lyra@ufrn.br (C.d.O.L.); roncalli@terra.com.br (A.G.R.d.C.O.); 2Federal Institute of Education, Science and Technology of Rio Grande do Norte, Natal 59015-300, RN, Brazil; eduarda.andrade@ifrn.edu.br; 3Postgraduate Program in Nutrition in Public Health, School of Public Health, University of São Paulo, São Paulo 05508-000, SP, Brazil; natiedantas@gmail.com; 4Department of Nutrition, Federal University of Rio Grande do Norte, Natal 59078-970, RN, Brazil; rab.andradebezerra@gmail.com

**Keywords:** adolescent, obesity, stunting, social determinants of health

## Abstract

(1) Background: The purpose of this study was to identify the prevalence of obesity and stunting among Brazilian adolescents and its associations with social determinants of health (individual, family, and school), grounded on the necessity of investigating the determinants of nutritional problems within this population. (2) Methods: A population-based survey was administered to 16,556 adolescents assessed by the 2015 National School Health Survey. Multivariate models of obesity and stunting were estimated from Multilevel Poisson Regressions. (3) Results: The prevalence of obesity among Brazilian adolescents (10.0%; 95% CI: 9.4–10.6) was associated directly with indifference or dissatisfaction with body image, with eating breakfast four or fewer days a week, living with up to four people in the household, studying in private schools, and being from the South region, and was inversely associated with being female, 15 years old or older, with having the highest nutritional risk eating pattern, dining at fast-food restaurants, and eating while watching television or studying. The prevalence of stunting (2.3%; 95% CI: 2.0–2.8) was directly associated with the age of 15 years or older, and inversely associated with the lower number of residents living in the household, maternal education—decreasing gradient from literate to college level education, studying in urban schools, and being from the South and Central-West regions. (4) Conclusions: Obesity in adolescence presented behavioral determinants. Stunting and obesity have structural social determinants related, respectively, to worse and better socioeconomic position among Brazilian adolescents.

## 1. Introduction

The prevalence of obesity and malnutrition coexists in various geographical areas globally [1]. These problems affect different age groups and are part of the challenges and issues of nutrition in adolescence, according to a publication of the World Health Organization (WHO), which highlighted psychological and eating practices (eating patterns often commonly seen in adolescence, eating disorders, and cultural patterns) as well as socioeconomic factors linked to food availability and access (higher availability of ultra-processed food, lack of safe and nutritious food, food supplying issues, and poverty) as determinants of such inadequacies [2].

Obesity is a complex chronic condition described as the result of interactions between an obesogenic environment, epigenetic factors, stress, sedentary lifestyle, genetic inheritance, growth and development, diabetes, and mood disorders [3]. A significant increase in obesity has been identified worldwide in recent years, reaching 5.0% among children and adolescents and 12.0% among adults in 2015 [4,5]. This increase in prevalence was approximately 47.1% for childhood obesity in the last three decades. As no country has been able to reduce this growth in obesity prevalence, obesity-preventing actions and proper management focused on this population are essential [6]. The prevalence of obesity—a chronic condition that is associated with a higher metabolic risk—increased between 1974 and 1975 (1.1%) [7], 2008 and 2009 (4.9%) [8] and 2013 and 2014 (8.4%) [9] in the Brazilian adolescent population.

Stunting—an indicator of chronic malnutrition—in this population decreased for both genders between 2009 and 2015, reducing from 3.2% to 2.5% among boys and from 2.2% to 1.8% among girls [10]. It is known that this last nutritional deviation is related to worse living conditions, assessed by family indicators related to economic factors [11] and schooling [12] in adolescents in Brazil. Other studies name the following as height-deficit nutritional determinants among adolescents: fathers’ occupation and education level, family income, number of family members with an income, geographic place of residence, ethnicity, and nutritional knowledge [13]. It is evident that the height deficit is a marker of social vulnerability and food and nutrition insecurity as a result of interactions that are dependent on age, sex, and several other variables, such as genetic, environmental, dietetic, socioeconomic, developmental, behavioral, nutritional, metabolic, biochemical, and hormonal factors [14,15].

The recognition and constant updating of the prevalence of these nutritional problems (obesity and stunting), as markers of food and nutrition insecurity among adolescents, are essential to the surveillance of the food and nutritional care of this population [16]. These measures are part of a set of fundamental efforts to understand the health situation in adolescence, which is marked by vulnerability related to deficits, risks, and violence. Understanding this reality may result in the development of a health system that is more responsive to adolescents’ health problems [17,18].

In this context, the study of the social determinants of health (sociopolitical contexts, material circumstances, social positions, and behaviors) of nutritional disorders is a necessary action to generate relevant information for the process of promoting human rights and reducing social inequities, which are essential to increase food and nutrition security and promote adequate and healthy eating in adolescence [19,20]. During this stage of life, these social determinants are distributed in the different contexts in which adolescents are inserted, such as families, schools, and communities. Therefore, this study aimed to identify the prevalence of obesity and stunting among Brazilian adolescents and its associations with social determinants of health (individual, family, and school).

## 2. Methods

### 2.1. Study Design and Data Source

This study employed a population-based survey that used data from “Sample 2” of the 2015 National School Health Survey (PeNSE 2015), which collected information from Brazilian adolescents on individual aspects and their schools. The PeNSE was approved by the National Research Ethics Committee under registration no. 1.006.467 [21]. The present study meets the research ethics issues established in the National Health Council Resolution no. 510/2016 [22] by using secondary databases.

The 2015 PeNSE was conducted by the Brazilian Institute of Geography and Statistics (IBGE) and the Brazilian Ministry of Health, and its “Sample 2”, with complex sampling plan, evaluated 16,608 students from the 6th grade of primary schools to the 3rd grade of high school, from three shifts (morning, afternoon, and evening), who were enrolled in public and private schools, from urban and rural areas throughout Brazil. This study used a cluster sampling plan. The schools were divided into geographic extracts (primary sample unit) and then by cohort (secondary sample unit) through drawing. All students in the cohorts were invited to participate in the survey. The sample size in each extract considered the following parameters: approximate sampling error of 3% in absolute values for the estimate of 50%, confidence interval of 95%, and mean effect of the sampling plan equivalent to “3” in the first stage. The research design enabled the estimation of population parameters for each of the five macro-regions of the country (North, Northeast, Southeast, South, and Central-West) and, therefore, for Brazil. The collection of individual data was performed through a self-administered electronic questionnaire; the collection of data from the school, with a questionnaire applied to the person in charge, was conducted between April and September 2015. More information on the sampling design and other aspects of the research can be found in the publication of PeNSE 2015 [21].

In this 2015 enquiry, “Sample 2” was planned for being able to generate estimates to be compared to other international studies focused on Brazilian adolescents. “Sample 1” only included the evaluation of adolescents in the ninth year of middle school in the 2009 and 2012 PeNSE. In this study, we analyzed data from “Sample 2” as it was considered broader data in terms of the age of the participants, thus generating more reliable estimates of the adolescent population. The analyses described below utilized individual data from 16,556 adolescents, including information on weight, height, sex, and age available in the IBGE database. Regarding the total of students available in the original database, the study lost no more than 30 individuals, or 0.18%.

### 2.2. Study Variables

The dependent variables in this study were obesity and stunting, being classified as dichotomous. Both outcomes were estimated from sex, age, weight, and height data. Anthropometric measurements were collected by trained surveyors using portable electronic scales and a stadiometer, following guidelines on measurement in a private environment, followed by the completion of the self-applied PeNSE 2015 questionnaire. Anthropometric data were not collected from students who declined to participate in the process or from adolescents who presented any impairment that could hinder the anthropometry. Those responsible for collecting weight and height data took two measurements, and when these were different, they took a third measurement. However, only one measurement for each variable was recorded in the student’s data [21].

The definition of obesity and stunting among adolescents was based on the variables of age, gender, weight, height, and body mass index (BMI), which was calculated using the formula (weight (kg)/height^2^ (m)). These were processed in the AnthroPlus [23] software from the World Health Organization to obtain the z-scores for each adolescent. The growth standard proposed by the WHO [24] for BMI-for-age and height-for-age was adopted as a reference. The nutritional status variables were defined based on cutoff points from 10 to 19 years, 11 months, and 29 days [25]. Adolescents with a BMI-for-age higher than z-score +2 were classified as obese, and those with height-for-age lower than z-score −2 were classified as stunted (low or very low height for age). The final categories for stunting and obesity were (0) none, and (1) present.

The independent variables were representative of dimensions of the social determinants in health adapted for food and nutrition issues in adolescence [26], considering:The school’s sociopolitical and economic context: geographic macro-region (North, Northeast, Southeast, South, and Central-West).The school’s material circumstances: school situation (urban, rural), administrative dependence (public, private), food environment—canteen (none, present), alternative points for food purchase (none, present), and school garden (none, present).The socioeconomic position and material circumstances of the individual and family: gender (male, female), ethnicity (white, non-white—composed by the union of “black”, “mixed”, “indigenous”, and “East Asian”), age (10–14 years, 15–19 years), job (yes, no), maternal education level (uneducated, literate, primary school, high school, college, or did not know); number of residents in the household (≥5 residents, <5 residents).Individuals’ behavioral and psychosocial health factors: dietary pattern (higher nutritional risk, lower nutritional risk), breakfast consumption, lunch or dinner consumption with parents or caregivers, food consumption while watching TV or studying (regular ≥ 5 days, irregular < 5 days), having been to fast-food restaurants in the week before the survey (no, yes), practicing physical activity (<300 min/week, ≥300 min/week), and body satisfaction (satisfied, dissatisfied, or indifferent).

Dietary patterns were estimated from seven variables of weekly consumption of food groups using cluster analysis with the k-means procedure (non-hierarchical). The pattern of lower nutritional risk was characterized by higher consumption of beans, fruits, vegetables, and greens, and lower consumption of ultra-processed salty snacks, fried snacks, sweets, and soft drinks. The pattern of higher nutritional risk was characterized by lower frequency of weekly consumption of beans, fruits, vegetables, and greens, and higher weekly frequency for the other food markers. Additional information can be identified in the methodological study of Vale et al. [27].

### 2.3. Statistical Analyses

The analysis of obesity and stunting among adolescents was conducted considering the individual- and school-level independent variables representative of social determinants of health. Descriptive analysis was carried out to estimate the prevalence of the dependent variables for each independent variable, with their respective 95% confidence intervals (95% CI). Pearson’s chi-square test was used to identify significant differences (*p* < 0.05) of the prevalence of dependent variables between the categories of each independent variable.

Multilevel Poisson regression techniques, which allow the consideration of contextual effects on the independent variables and outcome, were used to estimate the obesity and stunting explicative models. This analysis starts with the estimation of a random intercept model (null model) to verify the possibility of a multilevel analysis for the outcomes, considering the observation if each dependent variable had a different distribution according to the school level. Once the viability was verified, bivariate Poisson multilevel regressions were executed, including each independent variable into the null model, to estimate crude prevalence ratios (PRc) and their 95% CIs considering the effect of the school context.

Subsequently, the individual and school level variables that displayed *p* < 0.20 were tested in multiple models. Model 1 was estimated using only the individual level variables in the bivariate analysis and only those with statistical significance (*p* < 0.05) were included. Model 2 was composed of school level variables significant in the bivariate analysis, combined with the independent variables from the final model 1. These models generated adjusted prevalence ratios (PRa) and 95% CI and the quality of the models was assessed by estimating and observing the change in variance and significance of the likelihood-ratio test (LRtest). The final models (model 2) were achieved when the variables were added and model 2 had variance reduction and remained significant (*p* < 0.05). The mitigating and accentuating effects of the insertion of the context variables were checked in the multilevel analysis on the individual variables’ PR.

All analyses were conducted using the Stata software version 13.0 (StataCorp LP., College Station, TX, USA) and only the descriptive analysis considered the sample design of the research, since in the PeNSE database sample weights for school level are not available.

## 3. Results

The Brazilian adolescent population studied was predominantly male, of non-white ethnicity, aged 15–19 years, living in households with up to four people, studying in public schools and in the Southeast region. The obesity prevalence was 10.0% (95% CI:9.4–10.6), while the stunting prevalence was 2.3% (95% CI: 2.0–2.8) (Table 1).

A higher prevalence of obesity was observed among males (10.5%; 95% CI: 9.6–11.4), individuals up to 14 years of age (11.9%; 95% CI: 11.0–12.8), and those from the Southern region (12.6%; 95% CI: 11.4–13.9). A higher prevalence of stunting was identified in individuals of 15 years of age or older (3.6%; 95% CI: 2.9–4.3), and those from the North (3.9%; 95% CI: 2.8–5.4) and Northeast (2.9%; 95% CI: 2.2–4.0) regions (Table 1).

The bivariate analyses only excluded the variables “presence of a canteen” and “lunch or dinner with parents or caregivers” from the multiple modeling of obesity, which did not have a *p*-value < 0.200. The multiple modeling of stunting satisfied the statistical criteria for the following variables: sex, age, lunch or dinner with parents or caregivers, body satisfaction, physical activity, maternal education level, number of residents, school status, administrative dependency, region, and presence of a canteen (Table 2).

In the final multilevel model, the prevalence of obesity was associated directly with indifference (PR = 2.47; 95% CI: 2.17–2.82) or dissatisfaction (PR = 3.36; 95% CI: 3.03–3.75) with body image, with eating breakfast four or fewer days a week (PR = 1.31; 95% CI: 1.19–1.44), living with up to four people in the household (PR = 1.23; 95% CI: 1.11–1.36), studying in private schools (PR = 1.16; 95% CI: 1.04–1.30), and being from the South region (PR = 1.22; 95% CI: 1.04–1.44). Obesity prevalence was inversely associated with being female (PR = 0.67; 95% CI: 0.61–0.73), being 15 years old or older (PR = 0.61; 95% CI: 0.55–0.68), having the highest nutritional risk eating pattern (PR = 0.72; 95% CI: 0.64–0.79), dining at fast-food restaurants (PR = 0.82; 95% CI: 0.74–0.90), and eating while watching television or studying (PR = 0.89; 95% CI: 0.81–0.99). This was identified in the multilevel model of obesity among Brazilian adolescents with better adjustment of PR and variance compared to the null model after adding the school context variables (Table 3).

Stunting was directly associated with the age of 15 years or older (PR = 3.61; 95% CI: 2.70–4.82), and inversely associated with the lower number of residents living in the household (PR = 0.68; 95% CI: 0.53–0.88), maternal education—decreasing gradient from literate (PR = 0.66; 95% CI: 0.42–1.04) to college education level (PR = 0.23; 95% CI: 0.13–0.42), studying in urban schools (PR = 0.52; 95% CI: 0.33–0.82), and being from the South (PR = 0.56; 95% CI: 0.36–0.86) and Central-West (PR = 0.47; 95% CI: 0.30–0.75) regions. There was a mitigating effect of the school context variables on the effect of the body satisfaction variable in this model of stunting. This indicated the importance of including school context factors to obtain this final model for stunting among Brazilian adolescents. The inclusion of the school variables allowed a better final model adjustment, while keeping its significance (Table 4).

## 4. Discussion

The results of this study support discussions on the polarization of nutritional problems in the context of social inequities in health experienced by the Brazilian population: obesity associated with structural determinants characterized by greater economic development and urbanization conditions, and spaces associated with individual eating behaviors [28,29,30]; and stunting, as a malnutrition and nutritional insecurity marker, associated with structural factors of higher social vulnerability and food insecurity [31].

The estimated prevalence of stunting (2.3%) for adolescents in Brazil is within the expected limit for an adolescent population (less than 2.3%) [24]. This marker of malnutrition and food insecurity shows significant values in adolescents with poorer social positions, lower material conditions, and less family income, such as those with mothers with no education (6.3%), from rural areas (5.6%), from the North region (3.9%), and those who are older (15–19 years) (3.6%).

The prevalence of stunting among Brazilian adolescents in 2015 was lower than the estimate for Brazilian adolescents in 2009 (2.9%). However, the associations were similar to those of that year, when the stunting deficit was more prevalent in males, in public schools, in adolescents whose mothers had less education, and in poorer families [12].

It is perceived that among adolescents from other countries, this nutritional problem is a result of social inequities and persistent food and nutrition insecurity proxy situations [16,32]. In this sense, this negative situation in Brazil is more prevalent among the population of the North and Northeast regions, which are territories with sociopolitical contexts marked by less access to many basic services, such as healthcare [33,34]. Reducing stunting in adolescents in these territories depends on actions such as cooperation between the government, non-governmental organizations, health teams, and communities (social control), aiming to include this age group in public health priorities. These actions to reduce stunting must be based on the structural improvement of food security and education for health and nutrition, and must prioritize adolescents, their families, and communities in rural regions and those living in more vulnerable socioeconomic contexts [35].

Concerning the prevalence of obesity among Brazilian adolescents (10.0%), the value identified in this study was higher than the estimates of 1974–1975 (1.1%) [7], 2008–2009 (4.9%) [8] and 2013–2014 (8.4%) [9]. This finding supports the growth trend of this nutritional condition in Brazil, as well as in the world, for which the 2015 estimate was 5.0% obesity among children and adolescents [4]. It should also be noted that the prevalence of this condition is progressively closer to the estimates for the adult population in the same year (12.0%) [5].

Actions to control this accelerated growth of obesity do not seem to have had the desired effects, even though they are priorities for science and public health [36]. In this context, the report “The Global Syndrome” highlights the need for double and triple actions based on changes in the food system to reduce obesity, malnutrition, and climate change. This would reduce the consumption of ultra-processed foods (obesity), the use of natural resources (climate change), and the cost of fresh meat (malnutrition) [1].

The higher obesity prevalence among Brazilian adolescents who are male, younger, and from a better socioeconomic position found in the present study is similar to the results from other studies developed in Brazil [12,37]. However, the inverse association with the two dietary practices of highest nutritional risk (higher nutritional risk eating pattern and dining at fast-food restaurants), and the direct association with behaviors of higher risk, such as dissatisfaction with body image and eating breakfast four or fewer days a week, should be discussed. These results connect adolescents with obesity to health-promoting eating practices (better eating pattern and not dining at fast-food restaurants), and with practices of higher nutritional risk (irregular breakfast consumption and higher body dissatisfaction), and highlight the inner complexity of the nutritional and health care of people with obesity. This situation demands urgent public health interventions at this stage of life, either for the treatment or prevention of obesity, to promote healthier behaviors considering its multifactorial character [3].

More frequent breakfast consumption among adolescents has been related to a variety of health benefits, such as lower body fat, lower fasting blood glucose levels, increased cardiorespiratory fitness, and a healthier cardiovascular profile [38,39]. According to the Dietary Guidelines for the Brazilian population [40], breakfast is one of the three main meals of the day and compared to snacks, this morning meal provides a higher intake of vitamins and minerals and lower intake of fats and cholesterol [41,42]. Therefore, it is an important marker of health and healthy eating at this period of life, which is also adopted by the Brazilian Food and Nutrition Surveillance System [25].

However, regular meals should be constantly promoted among this section of the public. It is evidenced that skipping meals, such as breakfast, is associated with a low-quality diet—low intake of fruit and vegetables and high sodium intake [43]. This finding confirms the importance of establishing a dietary routine with regular meals in adolescence to improve dietary intake, metabolic markers, and a healthy weight.

The higher frequencies of healthy eating practices (lower nutritional risk dietary pattern and not dining at fast-food restaurants) among adolescents with obesity attracted attention in the present study. Associations between higher adherence to better eating patterns and lower frequency of dining at fast-food restaurants have been described for adolescents in Brazil [44]. Difficulties of the association between worse adolescent eating patterns and anthropometric nutritional status were identified in a systematic review, which highlighted associations with metabolic risk factors such as total cholesterol, HDL-c, fasting glycemia, diastolic blood pressure, and metabolic syndrome, but not with obesity anthropometric markers [45]. A review of studies assessing adults showed a clearer association between ultra-processed food consumption and obesity, characterized as a moderate level of evidence [46].

One hypothesis on this issue is that in adolescence, food consumption aspects are not yet so expressive on body composition. These results can also be debated from the actions developed for adolescents with obesity, which may involve adolescents in this age group who are more concerned and who, therefore, change their eating patterns based on health recommendations and healthy eating. In practice, it is identified that adolescents who do not have an established nutritional disorder worry little about changing their diet and eating practices. Additional studies are needed to investigate these issues. New evidence regarding these relationships will allow adolescent food and nutrition to be worked on more assertively, developing actions to promote adolescent health and to prevent obesity [47].

The relationship between body dissatisfaction and obesity needs to be discussed in more detail. It is known that this dissatisfaction brings losses in various aspects of health, such as worsening of the obesity situation or even issues of psychological illness, such as the development of eating disorders [48]. The evaluation of body dissatisfaction becomes an important marker for the treatment of people with obesity [49]. The American Academy of Pediatrics emphasized promoting a positive body image and not encouraging body dissatisfaction as a strategy to motivate healthy eating habits in actions to prevent obesity and eating disorders among adolescents [50].

The assessment of body image satisfaction is recommended in the process of weight and height evaluation among Brazilian adolescents, as proposed by the publication “Protecting and caring for the health of adolescents in primary care” [51]. Considering these recommendations and the results of the present study, the inclusion of the variable in other protocols used as a reference by health professionals for the prevention and management of obesity in Brazil should be discussed, such as the Food and Nutritional Surveillance System [25], Basic Care Booklet No.38 [52], in manuals of the Brazilian Society of Pediatrics [53], and the Brazilian Association for the Study of Obesity and Metabolic Syndrome [54]. Such discussions should be disseminated to other countries with the aim of integrating them into the food and nutritional care of adolescents [47].

Thus, it is highlighted that nutritional problems (obesity and stunting) are associated with socioeconomic issues in Brazil, as well as in other low- and middle-income countries [16]. In Brazil, adolescents with better life conditions (living with up to four people in the household, studying in private schools, and living in the South region) are more susceptible to obesity, and those adolescents in socioeconomic vulnerable situations (more residents living in the household, low maternal education, studying in rural schools, living in the North and Northeast regions) are more inclined to have stunting. Considering this association with the social determinants of health (food and nutrition insecurity proxy situations), in order to overcome these nutritional problems in Brazil, it is essential to re-establish the Food Security Council and the Brazilian Food Security System [55]. The actions to prevent and treat obesity and malnutrition must start by monitoring adolescents and families without ignoring the need for structural political actions to reduce the social inequities and food and nutritional insecurity experienced by the population [47].

In parallel, surveillance actions and food and nutrition education must be qualified to act on food choices, routines and behaviors, and issues of dissatisfaction with body image from the precept of the development of problematization of the of the adolescents’ reality and autonomy. It is important to highlight that, in Brazil, there are initiatives such as the Dietary Guidelines for the Brazilian population [40], the Food and Nutrition Education Reference Framework [56], and the publication Protect and Care [51], as well as the National School Feeding Program (Resolution/FNDE No. 6, of 8 May 2020) [57] and the School Health Program [58], but these actions need to be qualified to address adolescence. In this context, with the publication of Law No. 13,666 of 16 May 2018, discussions about healthy eating in public and private schools are facilitated by inserting food and nutrition education as a cross-cutting theme in the school curriculum [59].

The results of this study suggest the need to invest in interventions in private schools as the focus of obesity prevention and management actions. Historically, Brazilian private schools are less encouraged to develop food and nutrition education programs or offer healthy meals to students, because they are not linked to the National School Feeding Program, which is only intended for public schools [57].

This study’s main limitations refer to the unavailability of variables in the PeNSE database that would allow the exploration of other social determinants of stunting in adolescents, such as basic sanitation and drinking water supply, which have been assessed in other research studies analyzing the subject [60]. Additionally, the geographic division into macro-regions does not allow the analysis of more specific geographic spaces within Brazil. However, even with these limitations, this study is valid and relevant for estimating the prevalence of nutritional deviations representative of the Brazilian adolescent population.

## 5. Conclusions

Stunting among Brazilian adolescents is associated with the social determinants of health markers of worse sociopolitical and economic contexts, socioeconomic position, and the material circumstances of individuals and their families, with all of these being food insecurity markers linked to low food availability and access. In another context, obesity among Brazilian adolescents is associated with better social positioning and various behavioral factors and psychosocial conditions that pose a greater risk to health, such as the habit of not eating breakfast and body dissatisfaction.

These results are important for discussions about the management and provision of care for adolescents with these nutritional problems, and for the development of effective public actions aimed at preventing such problems. It is noteworthy that, in Brazil, these actions should consider the social determinants of health, as well as food and nutrition insecurity. Additionally, this study’s results contribute to the qualification of food and nutritional surveillance by providing important information for the development of actions, programs, and policies for health promotion and the prevention of stunting and obesity among Brazilian adolescents.

## Figures and Tables

**Table 1 nutrients-14-02334-t001:** Distribution of obesity and stunting prevalence among Brazilian adolescents and individual- and school-level variables, National School Health Survey, 2015.

Variables	*n*	Obesity			Stunting		
		%	95% CI	*p*-Value	%	95% CI	*p*-Value
Individual Level							
Socioeconomic position and material circumstances of the individual and family							
Gender				0.094			0.656
Male	8287	10.5	9.6–11.4		2.3	1.7–2.9	
Female	8269	9.4	8.6–10.3		2.4	2.0–3.0	
Ethnicity				0.575			0.178
White	6575	10.2	9.3–11.2		2	1.4–2.7	
Non-white	9958	9.8	9.1–10.6		2.6	2.1–3.1	
Age				<0.001			<0.001
10–14 years	9400	11.9	11.0–12.8		0.9	0.7–1.3	
15–19 years	7156	8.3	7.5–9.2		3.6	2.9–4.3	
Maternal education level				0.017			<0.001
Uneducated	749	7.9	5.8–10.6		6.3	3.8–10.5	
Literate	2735	9.8	8.4–11.3		3.3	2.4–4.5	
Primary schools	2002	8.8	7.3–10.5		1.8	1.0–3.2	
High school	3769	9.3	8.1–10.7		1.5	1.0–2.2	
College	3099	12.3	10.7–14.1		1.1	0.6–2.1	
Did not know	4168	10.5	9.3–11.9		2.4	1.8–3.2	
Number of residents in the household				<0.001			0.012
≥5 residents	6361	8.6	7.7–9.6		3	2.3–3.8	
<5 residents	10,180	10.8	10.1–11.7		1.9	1.5–2.4	
Individual’s behavioral and psychosocial health factors							
Dietary pattern				<0.001			0.765
Lower nutritional risk	10,257	11.1	10.3–11.9		2.2	1.8–2.8	
Higher nutritional risk	6153	8.2	7.3–9.2		2.4	1.8–3.1	
Lunch or dinner consumption with parents or caregivers				0.693			0.882
Regular (≥5 days)	11,928	9.9	9.2–10.6		2.3	1.9–2.9	
Irregular (<5 days)	4603	10.2	9.0–11.4		2.4	1.8–3.2	
Food consumption while watching TV or studying				0.064			0.303
Regular (≥5 days)	9297	10.5	9.7–11.4		2.1	1.7–2.7	
Irregular (<5 days)	7244	9.3	8.5–10.3		2.6	2.0–3.2	
Breakfast consumption				0.064			0.303
Regular (≥5 days)	7244	9.3	8.5–10.3		2.6	2.0–3.2	
Irregular (<5 days)	9297	10.5	9.7–11.4		2.1	1.7–2.7	
Dining at fast-food restaurants				0.001			0.218
No	8715	10.9	10.0–11.8		2.1	1.7–2.6	
Yes	7804	8.8	8.0–9.7		2.6	2.0–3.4	
Body satisfaction				<0.001			0.066
Satisfied	11,528	6.5	5.9–7.1		2.5	2.1–3.0	
Indifferent	1807	14.8	12.8–1.0		3.1	1.8–5.2	
Dissatisfied	3040	20.6	18.7–22.6		1.4	0.9–2.2	
Physical activity practice				0.019			0.008
≥300 min/week	3433	11.4	10.0–13.0		1.4	1.0–2.1	
<300 min/week	13,013	9.6	8.9–10.3		2.6	2.1–3.1	
School Level							
School’s material circumstances							
School situation				0.063			<0.001
Rural	851	7.2	5.0–10.3		5.6	3.3–9.4	
Urban	15,705	10.1	9.5–10.8		2.1	1.8–2.5	
Administrative dependence				<0.001			<0.001
Public	12,381	9.5	8.8–10.2		2.6	2.2–3.1	
Private	4175	13	11.6–14.5		0.5	0.2–0.9	
Canteen				0.004			<0.001
None	7253	9.1	8.3–10.0		3	2.4–3.7	
Present	9303	10.9	10.0–11.8		1.6	1.2–2.2	
Alternative points for food purchase				0.197			0.907
None	11,601	10.2	9.5–11.0		2.3	1.9–2.9	
Present	4955	9.4	8.4–10.5		2.4	1.8–3.1	
School Garden				0.728			0.701
Present	4091	9.8	8.6–11.1		2.5	1.7–3.6	
None	12,465	10	9.3–10.7		2.3	1.9–2.8	
School’s sociopolitical and economic context							
Geographic macro-region				<0.001			<0.001
North	3188	9.1	7.9–10.3		3.9	2.8–5.4	
Northeast	3465	8.1	7.1–9.3		2.9	2.2–4.0	
Southeast	3276	10.5	9.4–11.7		2	1.5–2.8	
South	3207	12.6	11.4–13.9		1.4	1.0–2.0	
Central-West	3420	10.5	9.3–11.8		1.3	0.8–2.1	
Brazil	16,556	10.0	9.4–10.6		2.3	2.0–2.8	

**Table 2 nutrients-14-02334-t002:** Bivariate associations of obesity and stunting among Brazilian adolescents and individual and school level variables, National School Health Survey, 2015.

Variables	Obesity			Stunting		
	PRc	95% CI	*p*-Value	PRc	95% CI	*p*-Value
Individual Level						
Socioeconomic position and material circumstances of the individual and family						
Gender						
Male	Ref			Ref		
Female	0.80	(0.72–0.88)	<0.001	1.19	(0.93–1.52)	0.177
Ethnicity						
White	Ref			Ref		
Non-white	0.91	(0.82–1.00)	0.045	1.15	(0.86–1.50)	0.320
Age						
10–14 years	Ref			Ref		
15–19 years	0.66	(0.60–0.73)	<0.001	3.53	(2.60–4.79)	<0.001
Maternal education level						
Uneducated	Ref			Ref		
Literate	1.15	(0.89–1.51)	0.288	0.67	(0.43–1.06)	0.090
Primary schools	1.06	(0.80–1.40)	0.678	0.35	(0.20–0.62)	<0.001
High school	1.11	(0.85–1.44)	0.435	0.34	(0.21–0.56)	<0.001
College	1.33	(1.03–1.74)	0.032	0.19	(0.10–0.35)	<0.001
Did not know	1.26	(0.98–1.63)	0.075	0.53	(0.34–0.83)	0.005
Number of residents in the household						
≥5 residents	Ref			Ref		
<5 residents	1.30	(1.18–1.43)	<0.001	0.64	(0.50–0.82)	0.001
Individual’s behavioral and psychosocial health factors						
Dietary pattern						
Lower nutritional risk	Ref			Ref		
Higher nutritional risk	0.71	(0.64–0.78)	<0.001	0.98	(0.76–1.27)	0.880
Lunch or dinner consumption with parents or caregivers						
Regular (≥5 days)	Ref			Ref		
Irregular (<5 days)	1.05	(0.95–1.17)	0.335	1.21	(0.93–1.59)	0.159
Food consumption while watching TV or studying						
Regular (≥5 days)	Ref			Ref		
Irregular (<5 days)	0.87	(0.79–0.96)	0.005	1.07	(0.83–1.37)	0.619
Breakfast consumption						
Regular (≥5 days)	Ref			Ref		
Irregular (<5 days)	1.38	(1.25–1.52)	<0.001	0.94	(0.73–1.21)	0.982
Dining at fast-food restaurants						
No	Ref					
Yes	0.77	(0.70–0.85)	<0.001	0.96	(0.75–1.24)	0.772
Body satisfaction						
Satisfied	Ref			Ref		
Indifferent	2.44	(2.14–2.78)	<0.001	1.05	(0.71–1.54)	0.819
Dissatisfied	3.11	(2.81–3.45)	<0.001	0.62	(0.42–0.92)	0.017
Physical activity practice						
≥300 min/week	Ref			Ref		
<300 min/week	0.90	(0.80–1.00)	0.053	1.32	(0.94–1.85)	0.104
School Level						
School’s material circumstances						
School situation						
Rural	Ref			Ref		
Urban	1.45	(1.11–1.91)	0.007	0.41	(0.25–0.68)	<0.001
Administrative dependence						
Public	Ref			Ref		
Private	1.35	(1.20–1.53)	<0.001	0.29	(0.18–0.47)	<0.001
Canteen						
None	Ref			Ref		
Present	1.27	(1.14–1.42)	<0.001	0.43	(0.31–0.58)	<0.001
Alternative points for food purchase						
None	Ref			Ref		
Present	0.87	(0.76–0.98)	0.025	1.11	(0.79–1.55)	0.560
School Garden						
Present	Ref			Ref		
None	1.09	(0.96–1.24)	0.200	1.04	(0.72–1.51)	0.814
School’s sociopolitical and economic context						
Geographic macro-region						
North	Ref			Ref		
Northeast	0.96	(0.80–1.15)	0.627	0.77	(0.50–1.19)	0.242
Southeast	1.19	(1.00–1.43)	0.053	0.63	(0.40–1.00)	0.048
South	1.38	(1.16–1.64)	<0.001	0.52	(0.32–0.83)	0.007
Central-West	1.09	(0.91–1.31)	0.336	0.38	(0.23–0.63)	<0.001

**Table 3 nutrients-14-02334-t003:** Multilevel regression of obesity among Brazilian adolescents and individual- and school-level variables, National School Health Survey, 2015.

Variables	Null Model	Model 1		Model 2	
		PR (CI 95%)	*p*-Value	PR (CI 95%)	*p*-Value
School Level					
Administrative dependence					
Private				1.16 (1.04–1.30)	0.010
Geographic macro-region					
Northeast				0.92 (0.78–1.10)	0.369
Southeast				1.10 (0.93–1.30)	0.265
South				1.22 (1.04–1.44)	0.014
Central-West				0.99 (0.84–1.16)	0.869
Individual Level					
Gender					
Female		0.66 (0.60–0.73)	<0.001	0.67 (0.61–0.73)	<0.001
Age					
15–20 years		0.61 (0.55–0.68)	<0.001	0.61 (0.55–0.68)	<0.001
Dietary pattern					
Higher nutritional risk		0.72 (0.65–0.80)	<0.001	0.72 (0.64–0.79)	<0.001
Food consumption while watching TV or studying					
Regular (≥5 days)		0.88 (0.80–0.97)	0.009	0.89 (0.81–0.99)	0.024
Breakfast consumption					
Irregular (<5 days)		1.32 (1.20–1.45)	<0.001	1.31 (1.19–1.44)	<0.001
Dining at fast-food restaurants					
Yes		0.83 (0.75–0.91)	<0.001	0.82 (0.74–0.90)	<0.001
Body satisfaction					
Indifferent		2.51 (2.20–2.86)	<0.001	2.47(2.17–2.82)	<0.001
Dissatisfied		3.42 (3.08–3.80)	<0.001	3.36 (3.03–3.75)	0.010
Number of residents in the household					
<5 residents		1.26 (1.14–1.39)	<0.001	1.23 (1.11–1.36)	<0.001
Fixed effects					
Intercept (CI 95%)	−2.244 (−2.304–(−2.185))	0.099 (0.087–0.112)		0.093(0.079–0.110)	
Random Effects	Variance (SE)	Variance (SE)		Variance (SE)	
Education level	0.092 (0.057–0.148)	0.042 (0.019–0.098)		0.033 (0.012–0.092)	
Variation (%)		−54.3		−64.1	
LR Test (Chi^2^; *p*-value)	30.32 (<0.001)	7.75 (*p* = 0.003)		5.84 (*p* = 0.007)	

**Table 4 nutrients-14-02334-t004:** Multilevel regression of stunting among Brazilian adolescents and individual- and school-level variables, National School Health Survey, 2015.

Variables	Null Model	Model 1		Model 2	
		PR (CI 95%)	*p*-Value	PR (CI 95%)	*p*-Value
School Level					
School situation					
Urban				0.52 (0.33–0.82)	0.005
Geographic macro-region					
Northeast				0.72 (0.49–1.05)	0.090
Southeast				0.70 (0.46–1.05)	0.082
South				0.56 (0.36–0.86)	0.008
Central-West				0.47 (0.30–0.75)	0.001
Individual Level					
Age					
15–20 years		3.71 (2.76–4.99)	<0.001	3.61 (2.70–4.82)	<0.001
Body satisfaction					
Indifferent		1.03 (0.69–1.52)	0.894		
Dissatisfied		0.61 (0.42–0.90)	<0.013		
Maternal education level					
Literate		0.59 (0.38–0.94)	0.026	0.66 (0.42–1.04)	0.075
Primary schools		0.34 (0.20–0.59)	<0.001	0.38 (0.22–0.66)	0.001
High school		0.33 (0.20–0.54)	<0.001	0.36 (0.22–0.60)	<0.001
College		0.21 (0.11–0.39)	<0.001	0.23 (0.13–0.42)	<0.001
Did not know		0.64 (0.41–1.00)	0.051	0.70 (0.44–1.10)	0.117
Number of residents in the household					
<5 residents		0.64 (0.50–0.83)	0.001	0.68 (0.53–0.88)	0.003
Fixed effects					
Intercept (CI 95%)	−4.474 (−4.679–(−4.269))	0.018 (0.011–0.030)		0.043(0.024–0.077)	
Random CI Effects	Variance (SE)	Variance (SE)		Variance (SE)	
Education level	0.649 (0.387–1.092)	0.306 (0.128–0.732)		0.197 (0.061–0.634)	
Variation (%)		−52.9		−69.6	
LR Test (Chi^2^; *p*-value)	29.75 (<0.001)	7.44 (*p* = 0.003)		3.73 (*p* = 0.027)	

## Data Availability

Not applicable.

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
