# Peer review of "Social Determinants of Obesity and Stunting among Brazilian Adolescents: A Multilevel Analysis"

_nutrients, 2022, doi:10.3390/nu14112334_

Round 1
Reviewer 1 Report
The manuscript addresses adolescent obesity and growth retardation, an issue of great relevance for Brazil and many other societies.Validity of the research is questioned as authors do not state that their sample was representative of the Brazilian adolescent population. Furthermore, no information is provided on the sampling procedure used; rather, one article in Spanish is cited for the reader to seek details.
Additional aspects of the study are inadequately touched upon, both in regards to the methodology followed and to the interpretation of the findings. One such aspects is the -unexpected- inverse relationship which was observed between risk of obesity and a "dietary pattern of higher nutritional risk", frequenting fast food outlets and having meals in front of the TV screen frequently.
Manuscript is not easy to understand as it is marked by linguistic flaws- lengthy and perplexed expressions, syntax errors, incorrect use of prepositions, unsuitable choice of words.
Author Response
Reviewer,
Thank you for your availability in reading our article. Your suggestions were considered by us for the improvement of our manuscript.
According to each suggestion, we have made the following corrections:
We added information about the validity of the research presenting more information about the sample that is representative of the Brazilian adolescent population. In addition, we provided more information about the sampling procedure used. With this, readers will be able to understand the methodology without having to read the documents in Portuguese.
We corrected the description of the methodology and the writing of some passages in "results", "discussion" and "conclusions" to improve the understanding of the findings of our study. We rewrote the passages about the "inverse (unexpected) relationship observed between obesity risk and "higher nutritional risk eating pattern", frequenting fast food establishments and having meals in front of the TV screen frequently". We hope that the modifications made are adequate to your notes.
The added excerpts and references are written in red.
We apologize for the linguistic flaws as English is not our original language. To correct this linguistic problem, the manuscript has been revised by an English language specialist.
Best regards,
Reviewer 2 Report
Very good work, nicely written and interesting.
I think the authors only need to improve the introduction about determinants of nutritional problems among adolescents and add some good references about that.
The following work is highly recommended to be cited and included in the reference list:
Body image dissatisfaction in individuals with obesity seeking bariatric surgery: Exploring the burden of new mediating factors. Bianciardi et al. Rivista di Psichiatria vol 54, issue 1, pages 8-17 2019
Author Response
Reviewer,
Thank you for your availability in reading our article. Your suggestions were considered by us for the improvement of our manuscript.
According to each suggestion, we have made the following corrections:
1- We have modified some parts of the introduction to highlight the determinants of nutritional problems among adolescents. We also added references on the determinants of obesity and stunting in accordance with your recommendations.
2- The article "Dissatisfaction with body image in obese individuals seeking bariatric surgery: exploring the burden of new mediating factors. Bianciardi et al. Rivista di Psichiatria vol 54, issue 1, pages 8-17 2019" was incorporated into the discussion and references list. This suggestion was important to further justify the importance of considering body dissatisfaction in the care of people with obesity.
The added excerpts and references are written in red.
We apologize for the linguistic flaws as English is not our original language. To correct this linguistic problem, the manuscript has been revised by an English language specialist.
Best regards,
Round 2
Reviewer 1 Report
A few aspects have been improved in the revised manuscript; this improvement mainly concerns the way in which results are presented in the text. However, serious weaknesses persist. Most importantly, contradictions are detected between findings reported in the Results Section and conclusions drawn in Discussion and Abstract; e.g. authors reached to the conclusion that "prevalence of obesity among Brazilian adolescents was -positively- associated with (....) NOT eating in fast food ", a conclusion that directly contradicts the results presented earlier in Table 3 and in accompanying text. Similarly, the conclusions drawn in regards to the role of the geographic location in predicting prevalence of obesity contradict the findings reported in the Results section.
Author Response
Reviewer,
Thank you for your availability to read our article again. Your suggestions were considered by us for the improvement of our manuscript. According to each suggestion, we made the corrections described below.
We have reviewed all the text to identify inconsistencies and differences between the results presented in the tables and in the topic "Results" and the information present in the "abstract", "discussion" and "conclusion". We corrected the inadequacies pointed out, such as the direct and inverse associations between obesity and stunting with the secondary variables. We corrected parts of the text and added some sentences to improve the understanding of the readers. All modifications of the text are written in red.
We apologize for the linguistic imperfections. The English language is not our original language. To solve the problem, the manuscript was again proofread and edited by an English language expert. Unfortunately, we were not able to hire the service suggested by Nutrients. The value is too high to be paid by the authors themselves (considering the devaluation of the Brazilian currency in the international scenario). Our University only pays for the more specific editing service when the article is approved. In case our article is considered of quality and is approved for publication, we may request payment for the language revision from our institution before publication.
We are available to make any other corrections that may be necessary.
Kind regards,